# A Comparative Clinical Study of Lateral Lumbar Interbody Fusion between Patients with Multiply Operated Back and Patients with First-Time Surgery

**DOI:** 10.3390/medicina59020342

**Published:** 2023-02-10

**Authors:** Masato Nakano, Hayato Futakawa, Shigeharu Nogami, Miho Kondo, Tatsuro Imai, Yoshiharu Kawaguchi

**Affiliations:** 1Department of Orthopaedic Surgery, Takaoka City Hospital, Toyama 933-8550, Japan; 2Department of Orthopaedic Surgery, University of Toyama, Toyama 930-0190, Japan

**Keywords:** multiply operated back, failed back spinal syndrome, lateral lumbar interbody fusion minimally invasive spinal stabilization, minimally invasive spinal treatment

## Abstract

*Background and Objectives:* Patients with multiply operated back (MOB) may suffer from persistent lower-back pain associated with back muscle damage and epidural fibrosis following repeated back surgery (Failed Back Spinal Syndrome). Lateral lumbar interbody fusion (LLIF) is considered to be favorable for MOB patients. However, little scientific research has been carried out for this issue due to variety of the pathogenesis. The purpose of this study was to compare the clinical results of MOB patients and first-time surgery patients who underwent LLIF for lumbar spinal stenosis with degenerative scoliosis and/or degenerative spondylolisthesis (spinal instability). *Materials and Methods:* LLIF was performed for lumbar spinal stenosis with spinal instability in 55 consecutive cases of single hospital (30 males, 25 females, average age: 69). Clinical outcomes were compared between the MOB patient group (MOB group) and the first surgical patient group (F group). We evaluated the Japanese Orthopaedic Association (JOA) scores and JOA Back Pain Evaluation Questionnaire (JOABPEQ) before and 2 years after surgery. MOB patients were defined as those who have had one or more lumbar surgery on the same intervertebral or adjacent vertebrae in the past. *Results:* There were 20 cases (12 males, 8 females, average age: 71) in MOB group, and 35 cases (19 males, 16 females, average age: 68) in the F group. There was no significant difference between the two groups in terms of age, sex, number of intervertebral fixations, modic change in fused intervertebral end plate, score of brief scale for evaluation of psychiatric problem, lumbar lordosis, and sagittal vertical axis before and after surgery. The preoperative JOA scores averaged 12.5 points in the MOB group and averaged 11.6 points in the F group. The postoperative JOA scores averaged 23.9 points in the MOB group and averaged 24.7 points in the F group. The preoperative JOABPEQ averaged 36.3 points in the MOB group and averaged 35.4 points in the F group. The postoperative average JOA score was 75.4 in the MOB group and 70.2 in the F group. *Conclusions:* Based on the results, there was no significant difference in clinical outcomes of the two groups, and there was no new residual lower-back pain. Thus, we considered that LLIF one option for patients with MOB.

## 1. Introduction

Patients with multiply operated back (MOB) may suffer from persistent lower-back pain associated with back muscle injury and epidural fibrosis after repeated back surgeries (failed back spinal syndrome; FBSS). FBSS has a wide variety of causes, including disc re-herniation, recurrence of spinal stenosis, postsurgical spinal instability and deformity, pseudoarthrosis, and adjacent segment disease, as well as iatrogenic factors [1,2,3,4,5,6,7]. The pathophysiology of FBSS remains unclear. Thus, there are various causes leading to MOB, and deeper knowledge and ability are required for pathological diagnosis. In addition, posterior lumbar reoperation for a MOB patient requires meticulous surgical planning and proficiency in the technique because complications including cerebrospinal fluid leakage, injury of the dura mater and the neural tissue are more likely to increase [3]. Anterior lumbar interbody fusion (ALIF) was considered advantageous for MOB patients in that: the ALIF approach through retroperitoneal virgin tissue could avoid the dissection of perineural scar tissue and para vertebral muscles; ALIF provided immediate discectomy, immobilization of the segmental motion of the intervertebral disc, and indirect neural decompression. However, few scientific studies have addressed the potential role for ALIF in the treatment of MOB patients. The previous noncontrolled case reports demonstrated that ALIF was a safe and effective procedure for the treatment of FBSS in selected patients [2,8,9]. Since the first reports in 2001 and 2006 [10,11], lateral lumbar interbody fusion (LLIF), which is the same retroperitoneal approach as ALIF, has been developed as a minimally invasive spinal surgery with useful devices [10,11,12,13]. Although LLIF was considered advantageous for MOB patients as well as ALIF, there have been few comparative studies on this issue due to the diversity of etiologies. The purpose of this study was to compare the clinical outcomes of MOB patients and first-time surgery patients who underwent LLIF for lumbar spinal stenosis with spinal instability.

## 2. Materials and Methods

A total of 55 patients have undergone LLIF and percutaneous pedicle screw fixation for lumbar spinal stenosis with degenerative scoliosis and/or degenerative spondylolisthesis since 2014 (Table 1). The average age at surgery was 69 years old (30 males, 25 females). LLIF was considered except in the cases needed for direct decompression. Patients with lumbar kyphosis that was needed for spinal posterior osteotomy were excluded from this study. MOB patients were defined as those who have had one or more lumbar surgery including posterior spinal decompression on the same intervertebral or adjacent vertebrae in the past. All LLIFs were performed by a single senior surgeon (MN).

### 2.1. Surgical Procedures

LLIF was performed under fluoroscopic imaging in a lateral decubitus position. LLIF including bone harvesting from the left iliac crest was performed through an approximately 3 cm skin incision per level. The dilators and retractor were inserted the psoas muscle using electromyography monitoring. A LLIF cage filled with autograft fragments and hydroxyapatite was inserted into each level. Percutaneous pedicle screws (PPS) and bent rods were inserted under C-arm fluoroscopic imaging and an electromyography monitoring guidance [14] in a prone position. The nerve root monitoring was carried out in all cases. The transforaminal interbody fusion (MIS-TLIF) was added when the fixation of L5/S1 was required (Table 1). The MIS-TLIF was added at either the symptomatic side or the concave side. After PPS screws were inserted to L5 and S1 pedicles, the L5/S1 discectomy was performed following unilateral facetectomy. A titanium-coated PEEK cage and autograft with hydroxyapatite granules were placed in the anterior part of the intervertebral disc space. Appropriate compressive force was applied between the adjacent PPSs.

### 2.2. Data Analysis

The clinical outcomes were compared between the MOB patient group (MOB group) and the first surgical patient group (F group) preoperatively and two years after surgery. We investigated demographic data including age, sex, number of the intervertebral fixation, and definitive diagnosis of etiology (Table 1). For psychogenic analysis, the Brief Scale for Psychiatric problems in Orthopaedic Patients (BS-POP) were recorded preoperatively [15]. Operation time and blood loss were investigated for the perioperative analysis related to surgical invasion. For radiologic analysis, the angle of lumbar lordosis (LL) and the sagittal vertebral axis (SVA) were measured before and at two years after surgery. On the preoperative magnetic resonance imaging (MRI), we evaluated existence of Modic type I or II changes. For clinical outcome associated with LBP, the pain domain of the Japanese Orthopaedic Association Back Pain Evaluation Questionnaire (JOABPEQ) score and the Japanese Orthopaedic Association scoring system (JOA score) were recorded before and at two years after surgery [16].

### 2.3. Statistical Analysis

For statistical analysis describing continuous variables and frequencies, means ± standard deviations were calculated. Either Fisher’s exact test, an independent *t*-test, or repeated measure ANOVA was conducted appropriately to compare the demographic data, preoperative scores, and radiographic variables before and at two years after surgery. A *p*-value < 0.05 was set as statistically significant in each test.

## 3. Results

There were 20 cases (males: 12 cases, females: 8 cases, average age: 71.4 ± 6.9 years old) in the MOB group, and 35 cases (males: 19 cases, females: 16 cases, average age: 67.7 ± 10.1 years old) in the F group (Table 1). There was no significant difference between the two groups in terms of age and sex. The average number of fixed intervertebral spaces was 2.0 ± 0.89 (ranged 1–3) in the MOB group and 2.0 ± 0.91 (ranged 1–4) in the F group. There was no significant difference in the proportion of cases in which L5/S1 fixation was added between the two groups (25% and 26%, respectively). Modic type I or II vertebral endplate degeneration at the fused segment was observed in 5 cases (25%) in the MOB group, and 12 (34%) in group F. The BS-POP in average was 17.1 (ranged 11–22) and 17.3 (ranged 10–26) points preoperatively in MOB group and F group, respectively, and showed no significant difference. The mean follow-up period was 53 months (ranged 24–96) and 51 months (ranged 24–84) in MOB group and F group, respectively (Table 2). The average operation time was 251 ± 82 min (ranged 135–420) and 253 ± 97 min (ranged 110–430), in MOB group and F group, respectively. Mean blood loss was 101 ± 103 mL (ranged 10–350) and 118 ± 113 mL (ranged 10–400), respectively. Mean LL was 30.4 degree and 30.2 degree before surgery and 33.5 degree and 30.2 degree at two years follow-up in MOB group and F group, respectively (Figure 1). Mean SVA was 54.6 mm and 56.3 mm before surgery, and 58.3 mm and 63.1 mm at follow-up (Figure 2). There were no differences in both groups between before and at two years after surgery in LL and SVA on the whole standing lateral radiograph (Figure 1 and Figure 2). All the spinal parameters showed no statistically significant difference between the two groups.

The preoperative JOABPEQ averaged 36.3 points in the MOB group and averaged 35.4 points in the F group. The postoperative average JOABPEQ was 75.4 in the MOB group and 70.2 in the F group. The preoperative JOA scores averaged 12.5 points in the MOB group, and averaged 11.6 points in the F group. The postoperative JOA scores averaged 23.9 points in the MOB group, averaged 24.7 points in the F group. In comparing the clinical outcomes, the JOABPEQ and the JOA scores were significantly improved after surgery compared with before surgery in both groups, but no significant difference was observed between the two groups (*p* = 0.474 in the JOABPEQ, *p* = 0.327 in the JOA scores) (Figure 3). No major complications were observed, and there were no characteristic complications concerning LLIF in each group (Table 2).

### Case Presentations

Case 1: Seventy-two-year-old woman with lumbar scoliosis and multiply operated back (Figure 4 and Figure 5). 

Two years and then one year before her first visit to our hospital, she underwent L4/5 and L2/3/4 posterior decompression surgery, respectively, at another hospital. Her right leg pain and the intermittent claudication gradually worsened, but conservative treatment was ineffective. The patient was referred to our hospital. Her chief complains were right leg pain, numbness and low back pain, causing gait disturbance. The JOA score was 10 points before surgery. Right L4 nerve block was effective but transient. There was L4/5 foraminal stenosis but no spinal central canal stenosis on MRI (Figure 4). The LLIF with percutaneous screw fixation at L4/5 level was performed. The operating time was 150 min, and the blood loss was 10 mL. The JOA score at two years follow-up was 25 points, and there were no neurological complications. The JOABPEQ score (Pain domain) was improved from 14 to 42.9. SVA remained unchanged during the follow-up period, whereas LL increased from 19.8 to 29.5. The bony fusion was successful two years after surgery with no residual pain (Figure 5). 

Case 2: Seventy-three-year-old man with lumbar spondylolisthesis following posterior decompression surgery (Figure 6). 

Since half a year ago, his bilateral sciatica had worsened, and the intermittent claudication had decreased to 200 m. He underwent L3/4 and L4/5 posterior decompression surgery at our hospital 6 years prior to the initial visit. His chief complaint was back pain and bilateral sciatica with no obvious muscle weakness at his initial visit. The preoperative JOA score was 13 points. L4 anterolisthesis of Grade 1 (slip ratio of 20%) was shown in the Lateral radiograph, and obvious lumbar stenosis at L4/5 was recognized on MRI. We performed L4/5 LLIF with PPS. The operation time was 170 min, and the blood loss was 10 mL. The slip ratio of L4/5 was completely reduced from 20% preoperatively to 0% postoperatively. Two years after surgery, there was no correction loss. His final JOA score was 28 points, and he had no neurological complications.

## 4. Discussion

The number of spinal surgeries is still increasing, and various surgical techniques and spinal instruments continue to be developed. One of the challenges is residual and recurring back pain after back surgery, known as FBSS [1,2,3,4,5,6,7]. Typical postoperative factors include lumbar re-stenosis, disc re-herniation, adjacent intervertebral disease, or postoperative instability. The incidence of FBSS after lumbar spinal surgery has been reported to range from 5% to 40% [1,4,6,7]. Patients with MOB may complain of persistent back pain associated with back muscle injury and epidural fibrosis after repeated back surgeries, and this condition was considered to be narrowly defined FBSS. However, many aspects of its pathology remain unclear, and psychological factors are often involved, making treatment extremely difficult. Therefore, its prevention is important.

Nowadays, minimally invasive spinal stabilization (MISt) has been applied to FBSS to prevent invasion to the back muscles and neural elements. MISt is a part of minimally invasive spine surgery that stabilizes the spine using minimally invasive spinal fusion techniques and dynamic stabilization techniques to correct imbalances caused by spinal instability, intervertebral instability, and spinal deformity [17,18]. MISt includes minimally invasive techniques—MIS-TLIF [19,20], LLIF including extreme lateral interbody fusion (XLIF) [10,11,12,13], oblique lumbar interbody fusion (OLIF) [21,22], etc. MISt has gained widespread recognition, especially with the development of PPS systems. The driving force behind the spread of MISt could be attributed to the widespread acceptance of methods that combine the use of PPS such as MIS-TLIF and XLIF/OLIF. Since the initial descriptions of ALIF in the 1930s and 1960 [23,24,25], ALIF has been recognized as one of the effective surgical procedures used to prevent FBSS, as well as to salvage surgery for FBSS. The mini-open retroperitoneal approach for ALIF was widely used since the late 1990s [26]. OLIF was also reported as a minimally invasive approach of ALIF in 1997 and 2012 [21,22]. LLIF, which was the same retroperitoneal approach as ALIF and OLIF, was first reported in 2001 and 2006 [10,11], and XLIF and OLIF have been developed as useful devices [10,11,12,13,21,22]. The previous study reported that XLIF could correct spinal deformities and indirectly decompress spinal stenosis in adult scoliosis [27]. Percutaneous reduction and fixation techniques with a combination of LLIF and PPS are becoming the standard procedure to reduce invasiveness and preserve the back muscles. Previous reports suggested that LLIF with PPS for degenerative spinal disorders such as scoliosis and spondylolisthesis could be expected to achieve clinical outcomes comparable to or better than MIS-TLIF [28,29,30]. A systematic review reported that indirect decompression with LLIF for degenerative spondylolisthesis was effective in significantly improving segmental lordosis and foraminal height and area compared with the posterior approach [31]. In the present study, LLIF with PPS was effective for lumbar spinal stenosis with spinal instability such as degenerative spondylolisthesis and degenerative scoliosis, as well as in the previous reports. From the results of this comparison between MOB patients and first-time surgery patients, there was no significant difference in clinical results, and there was no new residual low back pain, and similar improvement was obtained even with low back pain. Even with MOB, the same results as in the initial surgery group were obtained, and it was considered that LLIF surgery for MOB could be expected to have certain results. As for complications, no major complications including cerebrospinal fluid leaks, dural injury, or nerve injury were observed as with LLIF in MOB patients. At our suggestion, LLIF might be a minimally invasive technique that is gentle not only for the patient but also for the operator, especially in MOB patients.

There were limitations in the present study, as detailed below. First, this was a retrospective study consisting of non-randomized cohorts. Secondly, this surgery was performed in a single institution by a single surgeon (M.N.), although another co-authors (H.F. and S.N.) blindly evaluated the clinical outcomes and the radiographic data. Third, there was a bias in the selections of surgical procedure; either LLIF or LLIF combined with L5/S MIS-TLIF according to L5/S foraminal stenosis, instability, and L5/S neural decompression were required. In addition, there was variability in the pathological etiology between the two groups. Lastly, the number of MOB groups was relatively small. Therefore, further studies should continue to be considered to clarify comparative results between the subgroups.

## 5. Conclusions 

Clinical results from this comparison between MOB patients and first-time surgery patients showed no significant difference. There was no new residual low back pain in MOB patients as well as in first-time surgery patients. LLIF was considered as one of the surgical options for the patients with multiply operated back.

## Figures and Tables

**Figure 1 medicina-59-00342-f001:**
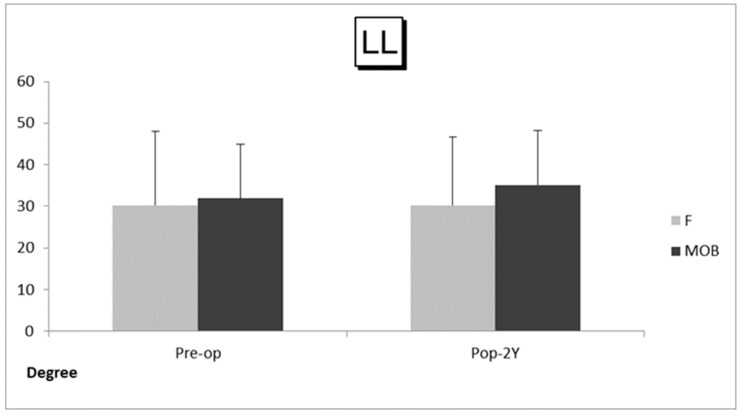
Angle of lumbar lordosis (LL) in the MOB patient group (MOB group) and the first surgical patient group (F group). The bars depict preoperative and follow-up average values and standard deviations in each group. Abbreviations: Pre-op, before surgery; Pop-2Y, at 2 years after surgery.

**Figure 2 medicina-59-00342-f002:**
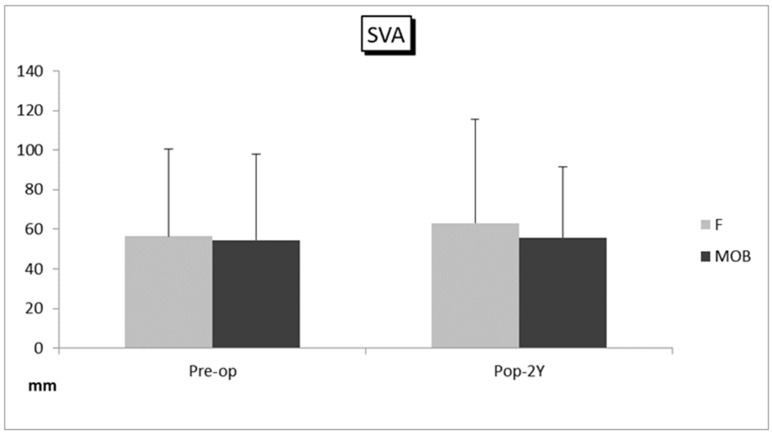
Sagittal vertebral axis (SVA) in the MOB patient group (MOB group) and the first surgical patient group (F group). The bars depict preoperative and follow-up average values and standard deviations in each group. Abbreviations: Pre-op, before surgery; Pop-2Y, at 2 years after surgery.

**Figure 3 medicina-59-00342-f003:**
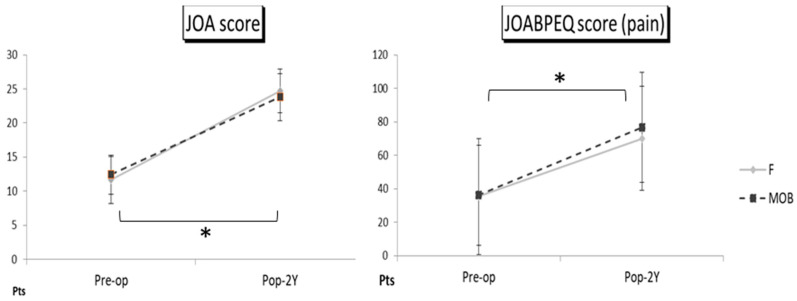
The JOA score and the JOABPEQ score at follow-up in the MOB patient group (MOB group) and the first surgical patient group (F group). The bars depict preoperative and follow-up average values and standard deviations in each group. The statistically significant difference was shown with an asterisk, and the postoperative level was compared with the preoperative level at *p* < 0.05. Comparing clinical outcomes, the JOABPEQ and the JOA scores were significantly improved at the two-year follow-up compared with the preoperative level in both groups, but there was no significant difference between the two groups.

**Figure 4 medicina-59-00342-f004:**
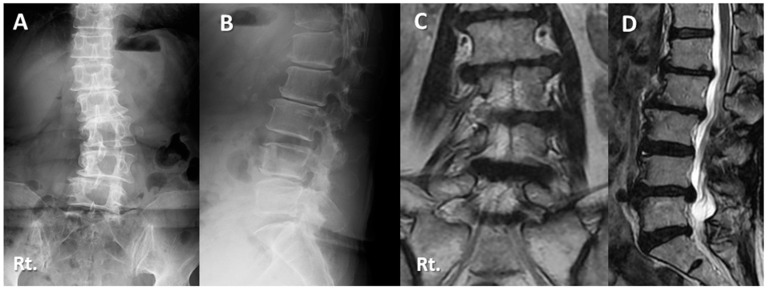
Seventy-two-year-old woman with lumbar degenerative scoliosis and multiply operated back: (**A**,**B**) Preoperative lateral radiographs showing lumbar degenerative scoliosis with postoperative state of L2/3/4/5 posterior decompression surgery. (**C**,**D**) Magnetic resonance imaging showed foraminal stenosis due to scoliosis. (**E**,**F**) Preoperative myelogram of functional lateral findings showing lumbar instability without spinal central canal stenosis. (**G**) Although she suffered from right (Rt.) leg pain and numbness at the L5 dermatome, causing significant disability of daily living and gait disturbance, the Rt. L5 nerve block was ineffective. (**H**) Rt. L4 nerve block was effective but transient.

**Figure 5 medicina-59-00342-f005:**
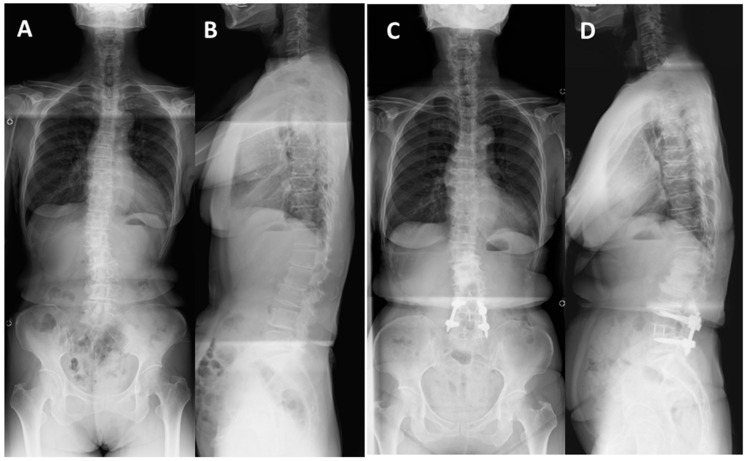
Seventy-two-year-old woman with lumbar degenerative scoliosis and multiply operated back: (**A**,**B**) Preoperative standing radiographs showing lumbar degenerative scoliosis. (**C**,**D**) Standing radiographs 5 year after surgery showed successful bony fusion and no correction loss at the L4/5 level that underwent LLIF. There were increased degenerative changes in the upper and middle lumbar spine, but no residual symptoms.

**Figure 6 medicina-59-00342-f006:**
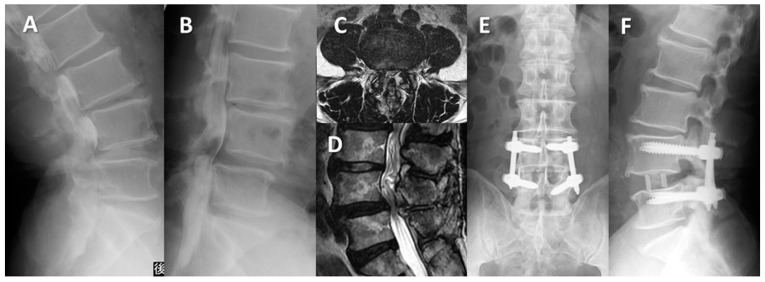
Seventy-three-year-old man with lumbar degenerative spondylolisthesis following posterior decompression surgery: (**A**,**B**) Preoperative myelogram lateral findings showing L4/L5 spondylolisthesis with 20% vertebra slippage and translation. (**C**,**D**) Magnetic resonance imaging showed severe stenosis at L4/L5. (**E**,**F**) Lateral radiograph two years after surgery showed no correction loss of slippage.

**Table 1 medicina-59-00342-t001:** Patient’s demographic data.

	MOB Group	F Group	*p*-Value
Patient number	20	35	
Average age at Surgery	71.4 ± 6.9 (56–83)	67.7 ± 10.1 (34–83)	0.221
Gender (Male:Female)	12:8	19:16	0.681
Average number of the fixed intervertebral disc {the proportion of cases in which L5/S1 fixation was added; %}	2.0 (1–3) {25%}	2.0 (1–4) {26%}	0.875 {0.447}
Disorders	Degenerative Spondylolisthesis: 7 * Degenerative Scoliosis: 9 * Degenerative Kypo-scoliosis: 5	Degenerative Spondylolisthesis: 13 Degenerative Scoliosis: 17 Degenerative Kypo-scoliosis: 7	
BS-POP	17.1 ± 2.7 (11–22)	17.3 ± 3.9 (10–26)	0.823
Modic type I or II vertebral endplate degeneration	5 (25%)	12 (34%)	0.799
Mean preoperative angle of lumbar lordosis (degree)	30.4 ± 11.8	30.2 ± 17.7	0.709
Mean preoperative sagittal vertebral axis (mm)	54.6 ± 46.6	56.3 ± 44.3	0.808
Preoperative JOABPEQ score (Pts)	36.3 ± 31.2	35.4 ± 34.9	0.941
Preoperative JOA score (Pts)	12.4 ± 2.9	11.6 ± 3.4	0.407

Abbreviations: BS-POP, the Brief Scale for Psychiatric problems in Orthopaedic Patients; JOABPEQ, the pain domain of Japanese Orthopaedic Association Back Pain Evaluation Questionnaire; JOA, Japan Orthopaedic Association; Pts, Points. *: Degenerative Spondylolisthesis and degenerative Scoliosis, including one case with the adjacent disease after posterior spinal fixation.

**Table 2 medicina-59-00342-t002:** Descriptive summarized results and complications in two groups.

	MOB Group	F Group	*p*-Value
Average follow-up period (month)	53 (24–96)	55 (24–84)	0.861
Average operation time (min)	251 (135–420)	253 (110–430)	0.830
Average estimated blood loss (ml)	101 (10–350)	118 (10–400)	0.464
**Complications**			
Major complication	0	0	
Revision surgery	0	1 (2.9%) *	
Minor complications	3 (15 %)	3 (8.6%)	
Superficial surgical site infection	1 (5%) **	0	
Transient LE pain, numbness, and/or weakness	1 (5%)	2 (5.7%)	
ALL rupture	0	1 (2.9%)	
Pseudarthrosis (all cases at L5/S1 level)	1 (5%)	1 (2.9%)*	

Abbreviations: LE, lower extremity; ALL, the anterior longitudinal ligament. *: Revision surgery in the F group was performed for pseudarthrosis of L5/S1 level at 4 years after L3/5 LLIF with L5/S MIS-TLIF. **: Superficial surgical site infection occurred the site of PPS and MIS-TLIF in the MOB group.

## Data Availability

Not applicable.

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
