# Peer review of "A Comparative Clinical Study of Lateral Lumbar Interbody Fusion between Patients with Multiply Operated Back and Patients with First-Time Surgery"

_medicina, 2023, doi:10.3390/medicina59020342_

Round 1
Reviewer 1 Report
Dear Authors,
I congratulate all the Authors for their contributions to the writing of the manuscript entitled “A Comparative Clinical Study of Lateral Lumbar Interbody Fusion between Patients with Multiple Operated Back and Patients with First-Time Surgery”, submitted for publication in the Medicina journal.
I have several comments on the manuscript:
1. Line 22, what is JOA? Japan Orthopaedic Association? Please clarify it when the abbreviation is first mentioned in the article.
2. Line 36, abbreviations do not need to be mentioned in the list of keywords.
3. Introduction – lacks references from other studies. The Authors should briefly and concisely discuss the background of the study topic. What are the differences between anterior spinal fusion and lateral lumbar interbody fusion, and how they can be utilized in MOB patients? What are the advantages of LLIF over the anterior spinal fusion? The Authors claimed these techniques are advantageous for MOB patients. Any references for this? Without a reference, the statement is highly speculative.
4. Table 1 & 2 – p-value should be written with p in italics.
5. Lines 80-91, abbreviations only need to be written when they are first mentioned in the manuscript. Please thoroughly check the manuscript for the correct use of abbreviations.
6. Line 99, “average 71 years old” – is this age mean? This can be written as 71.4±6.9 instead.
7. Line 139, the Authors claimed there was no significant difference between the two groups; what are the calculated p-values for these groups based on the JOABPEQ and JOA scores? This claim must be supported by a statistical analysis.
8. Line 200, what is XLIF and OLIF?
I consider the manuscript is sufficiently comprehensive, however it needs a thorough check for its grammar/use of punctuations. The introduction needs references where appropriate and must discuss the study background and/or other studies on the current issues/topic.
My sincere congratulations to all Authors.
Author Response
To Reviewer #1:
Reviewer #1’s comment: I congratulate all the Authors for their contributions to the writing of the manuscript entitled “A Comparative Clinical Study of Lateral Lumbar Interbody Fusion between Patients with Multiple Operated Back and Patients with First-Time Surgery”, submitted for publication in the Medicina journal.
Response: We are very thankful for the recommendations and we sincerely appreciate the time and effort you took to provide us with the insights.
Comment 1: Line 22, what is JOA? Japan Orthopaedic Association? Please clarify it when the abbreviation is first mentioned in the article.
Response: We thank the reviewer's comment. JOA means Japan Orthopaedic Association. We clarify it when the abbreviation is first mentioned in the article and the abstract.
Comment 2: Line 36, abbreviations do not need to be mentioned in the list of keywords.
Response: Thank you for your helpful comment. We revised the keywords.
Comment 3: Introduction – lacks references from other studies. The Authors should briefly and concisely discuss the background of the study topic. What are the differences between anterior spinal fusion and lateral lumbar interbody fusion, and how they can be utilized in MOB patients? What are the advantages of LLIF over the anterior spinal fusion? The Authors claimed these techniques are advantageous for MOB patients. Any references for this? Without a reference, the statement is highly speculative.
Response: We thank the reviewer for this kind suggestion. We significantly revised and added some texts about the background of the study topic and added references in Introduction and Reference part.
Comment 4: Table 1 & 2 – p-value should be written with p in italics.
Response: Thank you for your detailed check. We revised the Tables, Figure legend of Figure 3 and text in Statistical Analysis.
Comment 5: Lines 80-91, abbreviations only need to be written when they are first mentioned in the manuscript. Please thoroughly check the manuscript for the correct use of abbreviations.
Response: Thank you for your helpful comment. We thoroughly re-checked the manuscript for the correct use of abbreviations.
Comment 6: Line 99, “average 71 years old” – is this age mean? This can be written as 71.4±6.9 instead.
Response: Thank you for your helpful comment. We changed the text (Line 99-100) to “There were 20 cases (males: 12 cases, females: 8 cases, average age: 71.4 ± 6.9 years old) in the MOB group, and 35 cases (males: 19 cases, females: 16 cases, average age: 67.7 ± 10.1 years old) in the F group (Table 1)”
Comment 7: Line 139, the Authors claimed there was no significant difference between the two groups; what are the calculated p-values for these groups based on the JOABPEQ and JOA scores? This claim must be supported by a statistical analysis.
Response: Thank you for your valuable comment. We calculated an independent t-test to compared preoperative scores and postoperative scores respectively, and conducted repeated measure ANOVA to compared change of each score before and at two years after surgery in each score between the two groups. We added p-values in repeated measure ANOVA to the text (Line 139-140) as below: “In comparing the clinical outcomes, the JOABPEQ and the JOA scores were significantly improved after surgery compared to that before surgery in both groups, but no significant difference was observed between the two groups (p = 0.474 in the JOABPEQ, p = 0.327 in the JOA scores) (Figure 3).”
Comment 8: Line 200, what is XLIF and OLIF?
Response: Thank you for your helpful comment. We corrected use of abbreviations to “extreme lateral interbody fusion (XLIF) [7,8], oblique lumbar interbody fusion (OLIF) [18]” in Main text, Discussion, page 8, line 200.
Reviewer #1’s comment: I consider the manuscript is sufficiently comprehensive, however it needs a thorough check for its grammar/use of punctuations. The introduction needs references where appropriate and must discuss the study background and/or other studies on the current issues/topic.
My sincere congratulations to all Authors.
Response: Thank you very much for the positive feedback. We are very pleased to hear that you agreed with the importance of this report. According your suggestion, we re-checked and revised our manuscript.
Reviewer 2 Report
This study involved 55 consecutive cases undergoing LLIF for lumbar spinal stenosis with degenerative scoliosis and/or degenerative spondylolisthesis (spinal instability) to compare the clinical results of MOB patients and first-time surgery patients.
The authors evaluated JOA Back Pain Evaluation Questionnaire (JO-22 ABPEQ) and JOA scores before and 2 years after surgery between the MOB patient group (MOB group) and the first time operated patient group (F group).
No significant difference between the two groups regarding age, sex, number of intervertebral fixations, Modic change of fused intervertebral end plate, score of brief scale for evaluation of psychiatric problem, lumbar lordosis, and sagittal vertical axis before and after surgery. The preoperative JOA scores were 12.5 and 11.6 points in MOB group and F group respectively. The postoperative JOA scores were 23.9 and 24.7 points in MOB group and F group respectively. The preoperative JOABPEQ were 36.3 and 35.4 points in MOB group and F group respectively. The postoperative average JOA score was 75.4 in MOB group and 70.2 in the F group. No significant difference with respect to clinical outcomes was disclosed between two groups.
Even though this was a retrospective study consisting of non-randomized cohort, the subjective comparison of relevant parameters provided a significant finding clinically. By the setting of independent evaluation by another doctor who did not operate these patients, this study performed in a single institution by a single surgeon (M.N.) presented a fair comparison scientifically. Of course, further studies are needed to confirm the sustainability of the conclusion in the future.
Author Response
To Reviewer #2:
Reviewer’s comment:
This study involved 55 consecutive cases undergoing LLIF for lumbar spinal stenosis with degenerative scoliosis and/or degenerative spondylolisthesis (spinal instability) to compare the clinical results of MOB patients and first-time surgery patients.
The authors evaluated JOA Back Pain Evaluation Questionnaire (JO-22 ABPEQ) and JOA scores before and 2 years after surgery between the MOB patient group (MOB group) and the first time operated patient group (F group).
No significant difference between the two groups regarding age, sex, number of intervertebral fixations, Modic change of fused intervertebral end plate, score of brief scale for evaluation of psychiatric problem, lumbar lordosis, and sagittal vertical axis before and after surgery. The preoperative JOA scores were 12.5 and 11.6 points in MOB group and F group respectively. The postoperative JOA scores were 23.9 and 24.7 points in MOB group and F group respectively. The preoperative JOABPEQ were 36.3 and 35.4 points in MOB group and F group respectively. The postoperative average JOA score was 75.4 in MOB group and 70.2 in the F group. No significant difference with respect to clinical outcomes was disclosed between two groups.
Even though this was a retrospective study consisting of non-randomized cohort, the subjective comparison of relevant parameters provided a significant finding clinically. By the setting of independent evaluation by another doctor who did not operate these patients, this study performed in a single institution by a single surgeon (M.N.) presented a fair comparison scientifically. Of course, further studies are needed to confirm the sustainability of the conclusion in the future.
Response: We would like to thank the reviewer for taking the time to review our manuscript. We are pleased to know that you found our study interesting and agreed with the importance of this clinical study. According your important suggestion, we continue further studies to confirm the sustainability of the conclusion in the future.
Round 2
Reviewer 1 Report
Again, I congratulate all the Authors for their contributions to the writing of the manuscript.
I believe that the manuscript is now suitable for publication in the Medicina journal and can be accepted after spell checking/editing.